# The Analytical Solution of an Unsteady State Heat Transfer Model for the Confined Aquifer under the Influence of Water Temperature Variation in the River Channel

Ting Wei, Yuezan Tao *, Honglei Ren and Fei Lin

School of Civil Engineering, Hefei University of Technology, Hefei 230009, China
* Correspondence: taoyuezan@163.com

**Abstract:** The effect of water temperature variation in a river channel on groundwater temperature in the confined aquifer it cuts can be generalized to a one-dimensional thermal convection-conduction problem in which the boundary water temperature rises instantaneously and then remains constant. The basic equation of thermal transport for such a problem is the viscous Burgers equation, which is difficult to solve analytically. To solve this problem, the Cole–Hopf transform was used to convert the second-order nonlinear thermal convection-conduction equation into a heat conduction equation with exponential function-type boundary conditions. Considering the difficulty of calculating the inverse of the image function of the boundary function, the characteristics and properties of the Laplace transform were used to derive the theoretical solution of the model without relying on the transformation of the boundary function, and the analytical solution was obtained by substituting the boundary condition into the theoretical solution. The analytical solution was used to analyze the temperature response laws of aquifers to parameter variation. Subsequently, a 40-day numerical simulation was conducted to analyze the boundary influence range and the results from the analytical method were compared to those from the numerical method. The study shows that: (1) the greater the distance from the river canal and the lower the aquifer flow velocity, the slower the aquifer temperature changes; (2) the influence range of the river canal boundary increases from 18.19 m to 23.19 m at the end of simulation period as the groundwater seepage velocity $v$ increases from 0.08 m/d to 0.12 m/d; (3) the relative errors of the analytical and numerical methods are mostly less than 5%, confirming the rationality of the analytical solution.

**Keywords:** thermal convection-conduction model; Cole–Hopf transformation; Laplace transformation; analytical solution; FEFLOW

## 1. Introduction

The applied research that uses the heat transport equation in aquifers as the basic differential equation can be broadly divided into two categories: one is research on the utilization of shallow geothermal energy using aquifers and the groundwater endowed in them as thermal energy reservoirs [1], and the other is basic research related to the field of hydrogeology that utilizes groundwater temperature in aquifers as a tracer [2,3].

The primary goal of the first category of applications is to utilize shallow geothermal energy by performing heat exchange between aquifers and buildings through ground source heat pumps or groundwater source heat pumps. To study heat exchanger performance [4–6], water temperature variation laws in aquifers under the action of heat pump systems [7], etc., existing ground source heat pump and groundwater source heat pump systems, which rely on heat exchange boreholes and groundwater pumping-recharge wells, respectively, mostly establish corresponding mathematical models within a columnar coordinate system with the borehole as the origin.

The second category of application, i.e., the use of groundwater temperature as a tracer, has received wide attention in recent years in the study of river channel-groundwater

exchange due to the hydrothermal exchange between river channels and aquifers, as well as the sensitivity of temperature to hydraulic factors and the simplicity and low cost of temperature measurement [8]. These studies are typically conducted in a right-angle coordinate system with natural water bodies such as rivers and canals as boundaries to investigate, for example, the relationship between surface water and groundwater exchange [9,10], the law of recharge, runoff and discharge in groundwater systems [11], and the calculation of aquifer parameters [12], etc. Groundwater temperature is an important factor affecting groundwater flow patterns [13], groundwater chemistry and ecology [14], geothermal distribution [13], etc., but little literature studies the calculation of the effect of water temperature variation in the river channel on the groundwater temperature in the horizontal direction, which provides strong support for the estimation of aquifer parameters, joint management of rivers and aquifers, and a comprehensive understanding of hydrological processes.

For the above-mentioned heat transport problem in aquifers, the current main solving method is to make appropriate assumptions and generalizations on the problem and establish the corresponding mathematical model. The heat transport equation in aquifers—that is, the heat convection-conduction equation formed by groundwater seepage in conjunction with heat conduction in the solid skeleton of aquifers—is the fundamental differential equation in the above-mentioned model [15,16]. The heat convection-conduction equation belongs to the viscous Burgers' equation, which is extremely challenging to solve analytically [17,18]. Current studies are mostly based on numerical methods [17–22] and machine-learning methods [23,24], and the few analytical solutions are mostly under steady-state conditions [10,19]. However, the analytical method is an important tool for the study of mathematical physical models, which has the advantages of clear physical concepts, distinct physical meaning, and reliable theoretical bases. The analytical method not only reveals the intrinsic mechanisms and mathematical laws of the heat convection-conduction problem, which enables a more fundamental and in-depth recognition of the physical nature of the problem, but also, under specific boundary conditions, the problem can be solved more rapidly by the analytical method compared to the numerical and machine-learning methods, and the results obtained can be approximated as an exact solution, providing an effective verification of the applicability and correctness of other methods.

This study aims to give an analytical solution to the problem of water temperature variation in the aquifer due to the temperature variation in the river channel. To do so, a confined aquifer under the control of the river channel boundary was taken as an example and an unsteady one-dimensional thermal convection-conduction model in a semi-infinite domain under Dirichlet boundary conditions was established. The Cole–Hopf transformation was used to transform the thermal convection-conduction model into a heat conduction model in which the boundary of origin is an exponential function. The characteristics and properties of the Laplace transform were utilized to give the theoretical solution of this type of model, as it is difficult to solve the heat conduction model with an exponential function boundary directly using the integral transformation. The analytical solution was then obtained by combining it with the boundary conditions. Based on this, the variation laws of the aquifer temperature field under various parameters were studied in combination with mathematical examples. Subsequently, the river channel was generalized as a boundary with constant temperature, and the corresponding numerical simulation model based on FEFLOW software was established to study its influence range and to cross-validate the analytical solution (Figure 1).

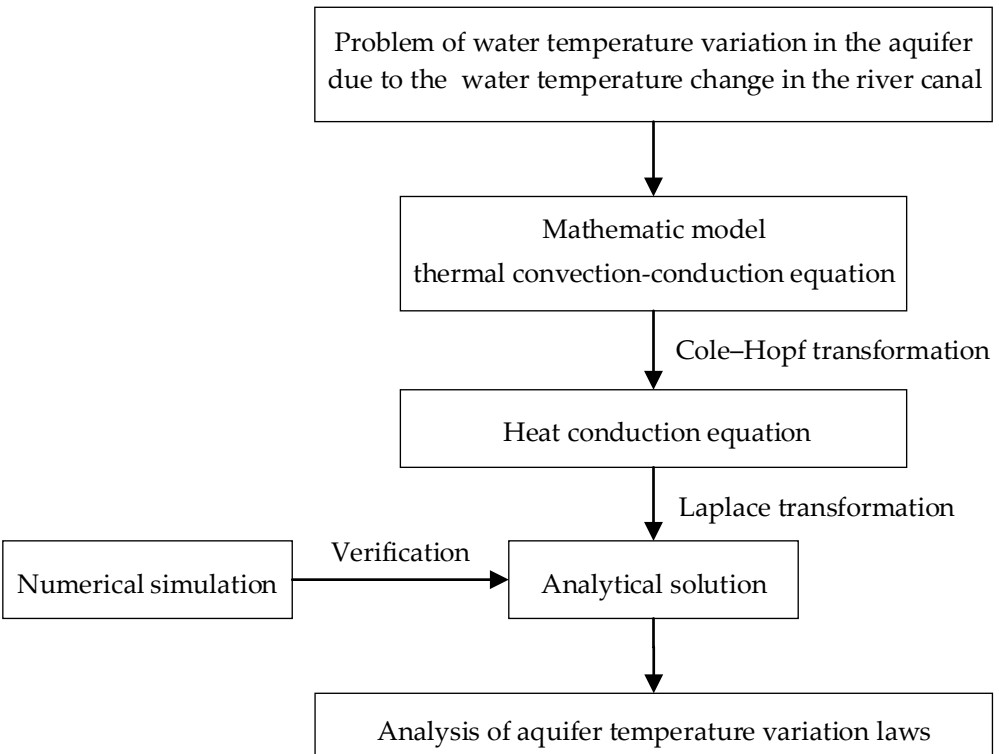

**Figure 1.** Flowchart of the research methodology.

　　This approach to solving problems, which is relatively straightforward and practical, can also be used to address issues in other fields where Burger's equation serves as the basic theoretical equation, such as convection-dispersion and convection-diffusion issues.

## 2. Mathematical Model and Its Analytical Solution

### 2.1. Mathematical Models

　　It is assumed that a horizontally spreading, homogeneous, and equally thick confined aquifer with a horizontal water barrier and groundwater flow is cut by a straight river canal (Figure 2). The aquifer's temperature field can therefore be simplified to a one-dimensional model in the horizontal direction, and the aquifer's temperature varies with time along the flow direction due to the effects of heat conduction and groundwater flow.

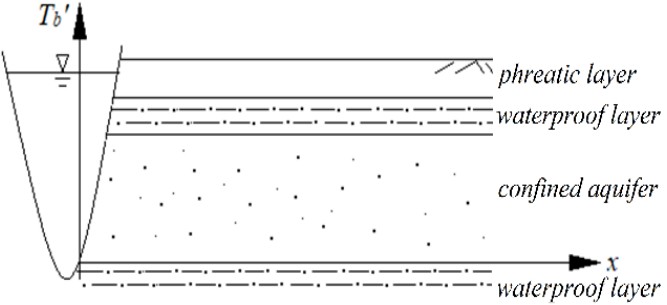

**Figure 2.** Schematic diagram of river canal and surrounding strata.

　　The water level of the river canal and the groundwater in the confined aquifer are in a stable state, and the groundwater flow velocity in the aquifer is a constant $v$. The water temperature of river canal water and aquifer are the same in the initial state as $T_b'(x,0)$, and the temperature at distance $x$ from the boundary at moment $t$ is $T_b'(x,t)$. Under the effect of industrial thermal discharge, the water temperature of the river canal rises instantaneously by $\Delta T_0$ and maintains this for a long time; thus, the river channel is considered as a

boundary with a constant temperature. The influence of the river channel on the water temperature of the aquifer is illustrated in Figure 3.

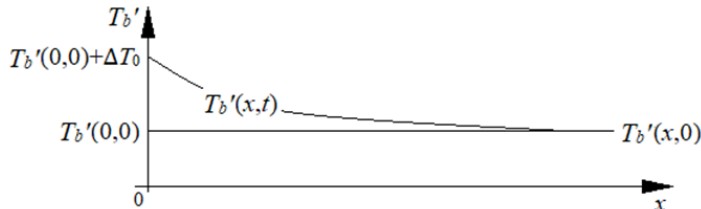

**Figure 3.** Schematic map of the effect of temperature variation in the river channel on the aquifer.

The mathematical model for the mentioned issue can be written as model (I) as follows, provided that the effect of the thermal conductivity of the upper and lower waterproof layers of the confined aquifer is not taken into account.

$$\frac{\partial T_b}{\partial t} = a\frac{\partial^2 T_b}{\partial x^2} - v\frac{\partial T_b}{\partial x} \qquad 0 < x < +\infty, t > 0 \tag{1}$$

$$T_b(x,t)|_{t=0} = 0 \qquad x > 0 \tag{2}$$

$$T_b(x,t)|_{x=0} = \Delta T_0 \quad t > 0 \tag{3}$$

$$T_b(x,t)|_{x\to\infty} = 0 \qquad t > 0 \tag{4}$$

where $t$ is the time (d); $x$ is the distance of the calculation point from the boundary (m); $T_b(x,t)$ is the excess temperature at distance $x$ from the boundary at moment $t$ (°C), i.e., $T_b(x,t) = T_b{}'(x,t) - T_b{}'(x,0)$, with the initial temperature as the reference; $a$ is the thermal diffusivity of the aquifer (m$^2$/d), which is a constant in the homogeneous aquifer; $v$ is the seepage velocity in the confined aquifer (m/d).

The generalized Equation (1) is a second-order nonlinear partial differential equation, which is difficult to solve directly. In this case, a function $\theta(x,t)$ can be established by Cole–Hopf transformation [18]: let $T(x,t) = T_b(x,t)/\theta(x,t)$, through which it is converted into a second-order linear partial differential equation.

By the Cole–Hopf transform [17,18], $\theta(x,t) = \exp[(vx/2a) - (v^2 t/4a)]$, we see that

$$T_b(x,t) = T(x,t)\cdot\theta(x,t) = T(x,t)\cdot\exp\left[(vx/2a) - \left(v^2 t/4a\right)\right] \tag{5}$$

Then the partial derivative of $T_b(x,t)$ with respect to $t$ is

$$\begin{aligned}\frac{\partial T_b(x,t)}{\partial t} &= \exp\left(\tfrac{vx}{2a}\right)\cdot\frac{\left[\frac{\partial T(x,t)}{\partial t}\cdot\exp\left(\frac{v^2 t}{4a}\right) - T(x,t)\cdot\exp\left(\frac{v^2 t}{4a}\right)\cdot\frac{v^2}{4a}\right]}{\left[\exp\left(\frac{v^2 t}{4a}\right)\right]^2}\\ &= \frac{\exp\left(\frac{vx}{2a}\right)}{\exp\left(\frac{v^2 t}{4a}\right)}\left[\frac{\partial T(x,t)}{\partial t} - T(x,t)\cdot\frac{v^2}{4a}\right]\end{aligned} \tag{6}$$

and the partial derivative of $T_b(x,t)$ with respect to $x$ is

$$\begin{aligned}\frac{\partial T_b(x,t)}{\partial x} &= \frac{1}{\exp\left(\frac{v^2 t}{4a}\right)}\left[\frac{\partial T(x,t)}{\partial x}\cdot\exp\left(\tfrac{vx}{2a}\right) + T(x,t)\cdot\exp\left(\tfrac{vx}{2a}\right)\cdot\frac{v}{2a}\right]\\ &= \frac{\exp\left(\frac{vx}{2a}\right)}{\exp\left(\frac{v^2 t}{4a}\right)}\left[\frac{\partial T(x,t)}{\partial x} + T(x,t)\cdot\frac{v}{2a}\right]\end{aligned} \tag{7}$$

Deriving the second partial derivative of $T_b(x,t)$ with respect to $x$ yields

$$\frac{\partial^2 T_b(x,t)}{\partial x^2} = \frac{\exp\left(\frac{vx}{2a}\right)}{\exp\left(\frac{v^2 t}{4a}\right)}\left[\frac{\partial T(x,t)}{\partial x} + T(x,t)\cdot\frac{v}{2a}\right]\cdot\frac{v}{2a} + \frac{\exp\left(\frac{vx}{2a}\right)}{\exp\left(\frac{v^2 t}{4a}\right)}\left[\frac{\partial^2 T(x,t)}{\partial x^2} + \frac{\partial T(x,t)}{\partial x}\cdot\frac{v}{2a}\right]$$

$$= \frac{\exp\left(\frac{vx}{2a}\right)}{\exp\left(\frac{v^2 t}{4a}\right)}\left[\frac{\partial T(x,t)}{\partial x}\cdot\frac{v}{a} + T(x,t)\cdot\frac{v^2}{4a^2} + \frac{\partial^2 T(x,t)}{\partial x^2}\right] \tag{8}$$

Substituting Equations (5)–(8) into model (I) yields model (II) as follow.

$$\frac{\partial T}{\partial t} = a\frac{\partial^2 T}{\partial x^2} \quad 0 < x < +\infty, t > 0 \tag{9}$$

$$T(x,t)|_{t=0} = 0 \qquad x > 0 \tag{10}$$

$$T(x,t)|_{x=0} = f(t) \quad t > 0 \tag{11}$$

$$T(x,t)|_{x\to\infty} = 0 \qquad t > 0 \tag{12}$$

where $f(t) = \Delta T_0/\theta(0,t) = \Delta T_0 \cdot \exp(v^2 t/4a)$.

In the process of converting the above heat convection-conduction model (I) containing second-order nonlinear partial differential equations, to the heat conduction model (II) containing second-order linear partial differential equations, the generalized equation in the model (I), i.e., the heat convection-conduction Equation (1), is converted to the heat conduction Equation (9), while the initial and boundary conditions (2)–(4) in the model (I) are converted into Equations (10)–(12).

It should be noted that the model in this paper ignores the influence of the vertical thermal conductivity of the upper and lower waterproof layers of the confined aquifer, which can be treated as the source-sink term of the generalized equation, but this will make the solution more complicated.

### 2.2. Theoretical Solution

In model (II), the boundary condition $f(t)$ at $x = 0$ is an exponential function, making it difficult to directly solve it using the integral transformation method. Therefore, the theoretical solution of this class of models is provided directly by the characteristics of the Laplace transformation, without relying on the integral transformation and inverse transformation processes of $T(x,t)$.

Applying the Laplace transformation on $t$ to model (II) yields model (III).

$$\frac{d^2\overline{T}}{dx^2} - \frac{s}{a}\overline{T} = 0 \tag{13}$$

$$\overline{T}(x,s)|_{x=0} = L[f(t)] \tag{14}$$

$$\overline{T}(x,s)|_{x\to\infty} = 0 \tag{15}$$

where $\overline{T}$ is the image function of the Laplace transformation of $T(x,t)$ with respect to $t$; $s$ is the Laplace operator; $L$ is the Laplace transformation operator.

When the boundary condition (11) is transformed into Equation (14), instead of solving its image function directly, only the operator is used during the operation.

The general solution of Equation (13) is

$$\overline{T}(x,s) = c_1\exp(\sqrt{\frac{s}{a}}\cdot x) + c_2\exp(-\sqrt{\frac{s}{a}}\cdot x) \tag{16}$$

where $c_1$ and $c_2$ are the constants to be determined.

From the definite solution conditions (14) and (15), the solution of model (III) is

$$\overline{T}(x,s) = L[f(t)]\cdot\exp(-\sqrt{\frac{s}{a}}\cdot x) \tag{17}$$

Instead of directly deriving the image function of $f(t)$, the boundary function $f(t)$ works in the form of an operator during the transformation, utilizing the properties of the Laplace transformation.

Performing the inverse Laplace transformation on Equation (17) and considering the convolution theorem of the Laplace inversion, we have

$$
\begin{aligned}
T(x,t) &= L^{-1}\left[\overline{T}(x,s)\right] \\
&= L^{-1}\left[L\left(f(t) \cdot \exp(-\sqrt{\tfrac{s}{a}} \cdot x)\right)\right] \\
&= L^{-1}[L(f(t))] * L^{-1}\left[\exp(-\sqrt{\tfrac{s}{a}} \cdot x)\right] \\
&= f(t) * L^{-1}\left[\exp(-\sqrt{\tfrac{s}{a}} \cdot x)\right]
\end{aligned}
\tag{18}
$$

where $L^{-1}$ is the inverse Laplace transformation operator; $*$ is the convolution operator.

Applying the inverse Laplace transformation to the complementary error function erfc($x/2\sqrt{at}$), gives

$$
L^{-1}\left[\frac{1}{s}\exp(-\sqrt{\frac{s}{a}} \cdot x)\right] = \mathrm{erfc}\left(\frac{x}{2\sqrt{at}}\right) = \frac{2}{\sqrt{\pi}}\int_{\frac{x}{2\sqrt{at}}}^{+\infty} e^{-\zeta^2}d\zeta
\tag{19}
$$

In order to apply Equations (18) and (19), it is necessary to further organize and expand Equation (18). From the differential property of the inverse Laplace transformation, it follows that

$$
\begin{aligned}
L^{-1}\left[\exp\left(-\sqrt{\tfrac{s}{a}} \cdot x\right)\right] &= L^{-1}\left\{s \cdot \left[\tfrac{1}{s} \cdot \exp(-\sqrt{\tfrac{s}{a}} \cdot x)\right]\right\} \\
&= \tfrac{d}{dt}\left\{L^{-1}\left[\tfrac{1}{s}\exp(-\sqrt{\tfrac{s}{a}} \cdot x)\right]\right\} \\
&= \frac{d\left[\mathrm{erfc}(x/2\sqrt{at})\right]}{dt}
\end{aligned}
\tag{20}
$$

Combining Equations (18) and (20) yields

$$
T(x,t) = f(t) * \frac{d\left[\mathrm{erfc}\left(\frac{x}{2\sqrt{at}}\right)\right]}{dt}
\tag{21}
$$

Utilizing the convolution differential property of the Laplace transformation, there is

$$
\begin{aligned}
&f(t) * \tfrac{d}{dt}\left[\mathrm{erfc}\left(\tfrac{x}{2\sqrt{at}}\right)\right] + f(t) \cdot \left[\mathrm{erfc}\left(\tfrac{x}{2\sqrt{at}}\right)\Big|_{t=0}\right] \\
&= \mathrm{erfc}\left(\tfrac{x}{2\sqrt{at}}\right) * \tfrac{d[f(t)]}{dt} + f(t)|_{t=0} \cdot \mathrm{erfc}\left(\tfrac{x}{2\sqrt{at}}\right)
\end{aligned}
\tag{22}
$$

Since $\mathrm{erfc}\left(x/2\sqrt{at}\right)\Big|_{t=0} = 0$, combining Equations (21) and (22) yields

$$
T(x,t) = f(t)|_{t=0} \cdot \mathrm{erfc}\left(\frac{x}{2\sqrt{at}}\right) + \mathrm{erfc}\left(\frac{x}{2\sqrt{at}}\right) * \frac{d[f(t)]}{dt}
\tag{23}
$$

Applying the exchange law of convolution, Equation (23) can be written in integral form as

$$
T(x,t) = f(t)|_{t=0} \cdot \mathrm{erfc}\left(\frac{x}{2\sqrt{at}}\right) + \int_0^t \frac{d[f(\zeta)]}{dt} \cdot \mathrm{erfc}\left(\frac{x}{2\sqrt{a(t-\zeta)}}\right)d\zeta
\tag{24}
$$

Substituting Equation (24) into Equation (5) yields

$$T_b(x,t) = [\int_0^t \frac{d[f(\zeta)]}{d\zeta} \cdot \text{erfc}\left(\frac{x}{2\sqrt{a(t-\zeta)}}\right) d\zeta + f(t)|_{t=0} \cdot \text{erfc}\left(\frac{x}{2\sqrt{at}}\right)] \cdot \frac{\exp\left(\frac{vx}{2a}\right)}{\exp\left(\frac{v^2t}{4a}\right)} \quad (25)$$

During the solving process, $f(t)$ is not directly involved in the transformation, so Equation (25) is the theoretical solution of the one-dimensional thermal convection-conduction model (I).

### 2.3. Analytical Solution

Substituting $f(t) = \Delta T_0 \cdot \exp(v^2t/4a)$ into Equation (25), gives

$$T_b(x,t) = \left\{\int_0^t \left[\frac{v^2}{4a}\exp\left(\frac{v^2\zeta}{4a}\right)\Delta T_0\right]\text{erfc}\left(\frac{x}{2\sqrt{a(t-\zeta)}}\right)d\zeta + \Delta T_0 \cdot \text{erfc}\left(\frac{x}{2\sqrt{at}}\right)\right\} \cdot \exp\left(\frac{vx}{2a} - \frac{v^2t}{4a}\right) \quad (26)$$

Equation (26) is the analytical solution of the thermal convection-conduction model (I) under the boundary condition of a constant temperature.

It is worth pointing out that related problems in other fields, such as the convection-diffusion model in atmospheric pollutants and the convection-dispersion model for solute transport in aquifers, have the same generalized equations as the thermal convection-conduction model in aquifers, and thus can also be solved by this method.

## 3. Materials and Methods

To verify the accuracy of the above analytical solution to the problem, i.e., Equation (26), a numerical simulation was conducted by combining an example.

### 3.1. Example Overview

As shown in Figure 2, the phreatic layer is 0.5 m thick and the aquifer is an 8 m-thick loose layer composed primarily of fine sand, with a waterproof layer 0.5 m above it and a waterproof layer 1.0 m below it. The main parameters of the aquifer are listed in Table 1.

**Table 1.** List of aquifer parameters.

| Parameter | Unit | Value |
|---|---|---|
| volume specific heat of solid skeleton, $c_s$ | J·(kg·°C)$^{-1}$ | $1.78 \times 10^3$ |
| volume specific heat of water, $c_w$ | J·(kg·°C)$^{-1}$ | $4.20 \times 10^3$ |
| density of solid skeleton, $\rho_s$ | kg·m$^{-3}$ | $1.50 \times 10^3$ |
| thermal conductivity of solid skeleton, $\lambda_s$ | W·(m·°C)$^{-1}$ | 1.92 |
| thermal conductivity of water, $\lambda_w$ | W·(m·°C)$^{-1}$ | 0.59 |
| porosity, $n$ | - | 0.40 |

The initial water temperature of the river canal and the aquifer is 18 °C. Under the effect of thermal discharge of power plants, the water temperature in the river channel rises rapidly to 22 °C in a short time and then maintains this for a long time, i.e., $\Delta T_0 = 4$ °C.

### 3.2. Numerical Simulation

A two-dimensional water-thermal coupling numerical simulation model was established using the finite element numerical simulation software FEFLOW [25], which is an interactive groundwater modeling program for 2D and 3D processes that are completely linked or uncoupled with thermo-hydro-chemical reactions in saturated or variable saturated systems [26]. The model was established as a steady-state flow and unsteady heat transfer model under saturated medium conditions, which is consistent with the settings of the analytical method, as well as boundary conditions, initial conditions, and parameter values. The model was 30 m long and 10 m high and was divided into four layers from

top to bottom in the height direction: phreatic layer, waterproof layer, confined aquifer, and waterproof layer. To reflect the temperature variation process and facilitate the subsequent comparison analysis with the results obtained by the analytical method, two virtual observation points were arranged at 0.5 m and 1 m from the boundary, respectively.

During the simulation, the system equations are solved using standard iterative methods based on FEFLOW. For the flow field (symmetric linear) equations group and the thermal transport (asymmetric linear) equations group, the Preconditioned Conjugate-Gradient method (PCG) and the Bi-Conjugate Gradient Stabilized method (BICGSTAB) were used for solving, respectively. Temporally, the Adams-Bash backward prediction/correction trapezoidal second order method (AB/TR) was utilized for the flow field and coupled flow field-thermal transport simulation. Spatially, the triangle grid generation technique was used to perform the spatial discretization in this simulation [27], which was based on the continuous Galerkin finite element method. The total model was divided into 1731 elements and 914 nodes (Figure 4), with local mesh refinement around virtual observation points.

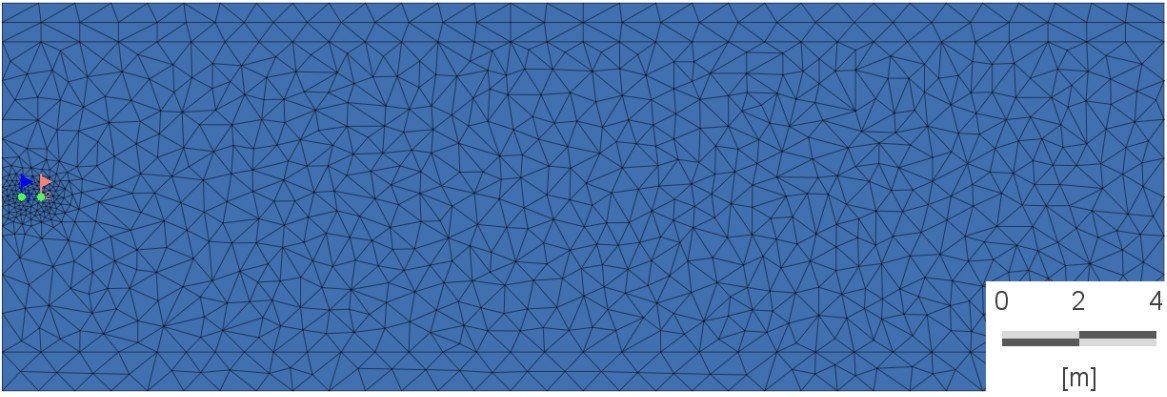

**Figure 4.** Mesh profile of the example.

The left and right sides of the aquifer were set as the given hydraulic head boundary, with the right side of the aquifer set to 9 m and the left side set to 9.79 m, 10 m, and 10.18 m (the trial runs of the model revealed that the corresponding groundwater flow velocity was stable at 0.08 m/d, 0.10 m/d, and 0.12 m/d respectively). Combined with the above example, the initial temperature of the aquifer was 18 °C, and the river channel on the left side was set as the constant temperature boundary, of which the value was assigned to 22 °C.

Although the numerical model is a flat two-dimensional model, it should be noted that the boundary is a straight river channel and the homogeneous aquifer is horizontally spreading. Therefore, the heat transfer process in the aquifer under the control of this river channel boundary is essentially a one-dimensional heat transfer problem of the vertical river channel.

## 4. Results

### 4.1. Analysis of Aquifer Temperature Variation Laws Basing on the Analytical Method

Based on the calculation formula for thermal diffusivity,

$$a = \frac{\lambda_w \times n + \lambda_s \times (1-n)}{[\rho_w \times n + \rho_s \times (1-n)] \times [c_w \times n + c_s \times (1-n)]} \tag{27}$$

the thermal diffusivity of the aquifer is calculated as $a = 1.4 \times 10^{-3}$ m$^2$/h = 0.0336 m$^2$/d, in combination with Table 1.

As mentioned above, $\Delta T_0 = 4$ °C, i.e., both $\Delta T_0$ and $a$ are fixed values for the aquifer. From Equation (26), it can be seen that the excess temperature $T_b(x,t)$ is related to the parameters $v$, $x$, and $t$. Therefore, one of the parameters can be set as a fixed value in turn,

and the variation laws of $T_b(x,t)$ caused by the variation of the other two parameters can be studied. It should be noted that Equation (26) contains a parametric variable proper integral, which can be calculated by substituting the corresponding values using WolframAlpha [28]. WolframAlpha is an online computational knowledge engine, which can output integration results and visualize the outcomes based on algorithms and data.

Figure 5 illustrates the variation of aquifer excess temperature $T_b(x,t)$ with time at various distances from the river canal at $v = 0.1$ m/d, $T_b(x,t)$ with time at various flow velocities at $x = 1$ m, and $T_b(x,t)$ with distance at different flow velocities at $t = 20$ d.

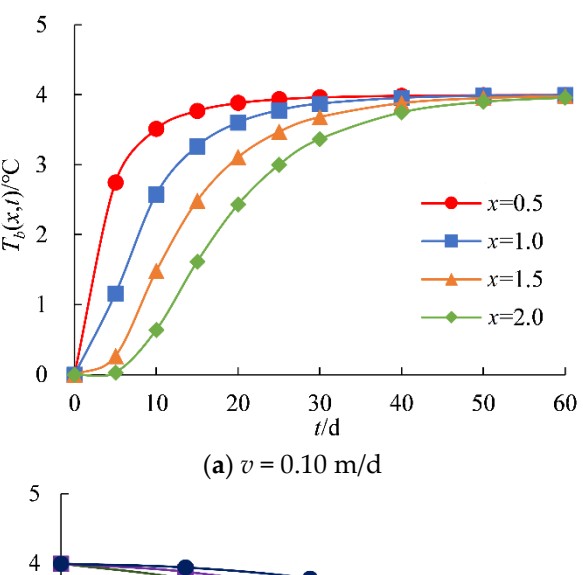

(**a**) $v = 0.10$ m/d

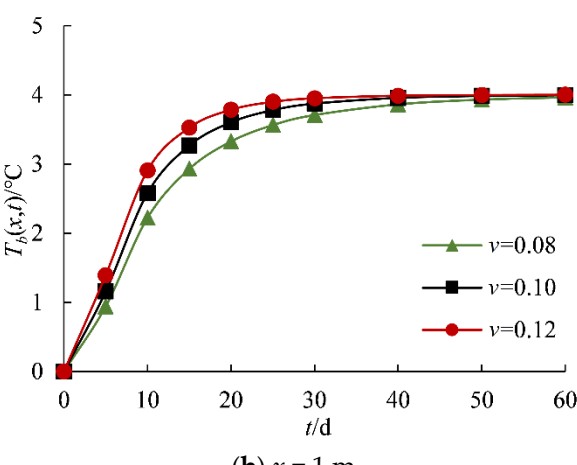

(**b**) $x = 1$ m

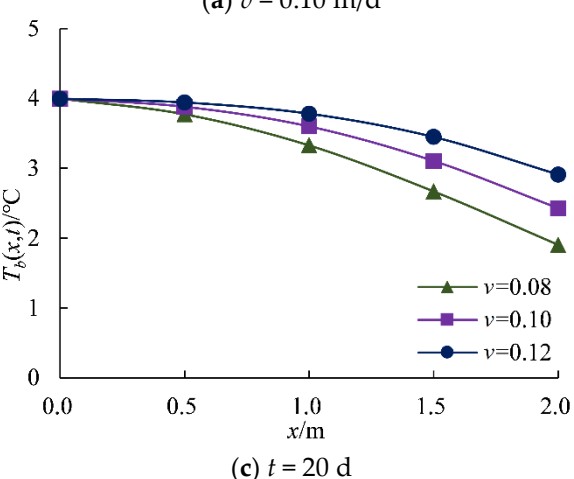

(**c**) $t = 20$ d

**Figure 5.** Variation curves of $T_b(x,t)$.

Figure 5 illustrates the results as follows:

(1)    at the same flow velocity, the further away from the river canal, the slower the aquifer temperature varies;

(2)    at the same distance from the river canal, the smaller the aquifer flow velocity, the slower the temperature changes;

(3)    at the same time, the larger the aquifer flow velocity, the smaller the temperature changes caused by the increased distance;

(4)    at the same distance from the river canal or at the same flow velocity, the aquifer temperature increases with time until it reaches a stable state.

### 4.2. Analysis of the Boundary Influence Range Basing on Numerical Simulation

Figures 6 and 7 depict the temperature distribution of the aquifer after 20 and 40 days of model operation, respectively.

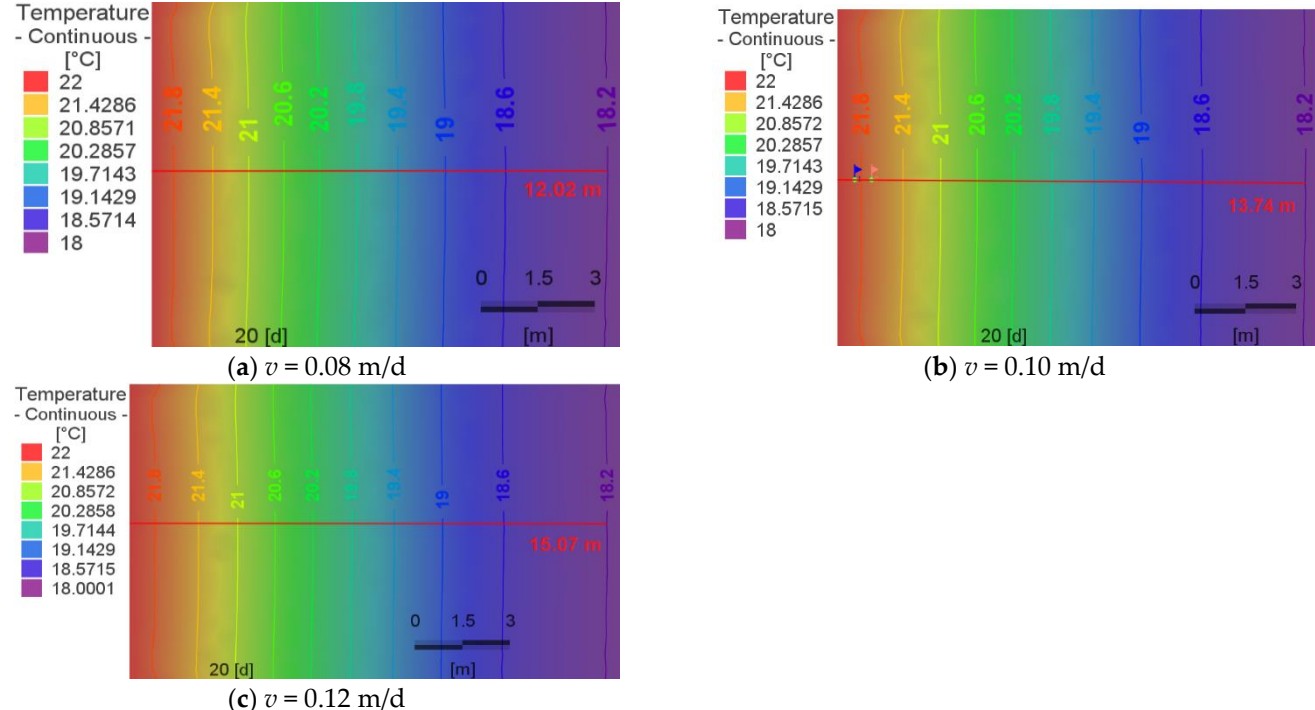

**Figure 6.** The temperature distribution of the aquifer after 20 days of model operation.

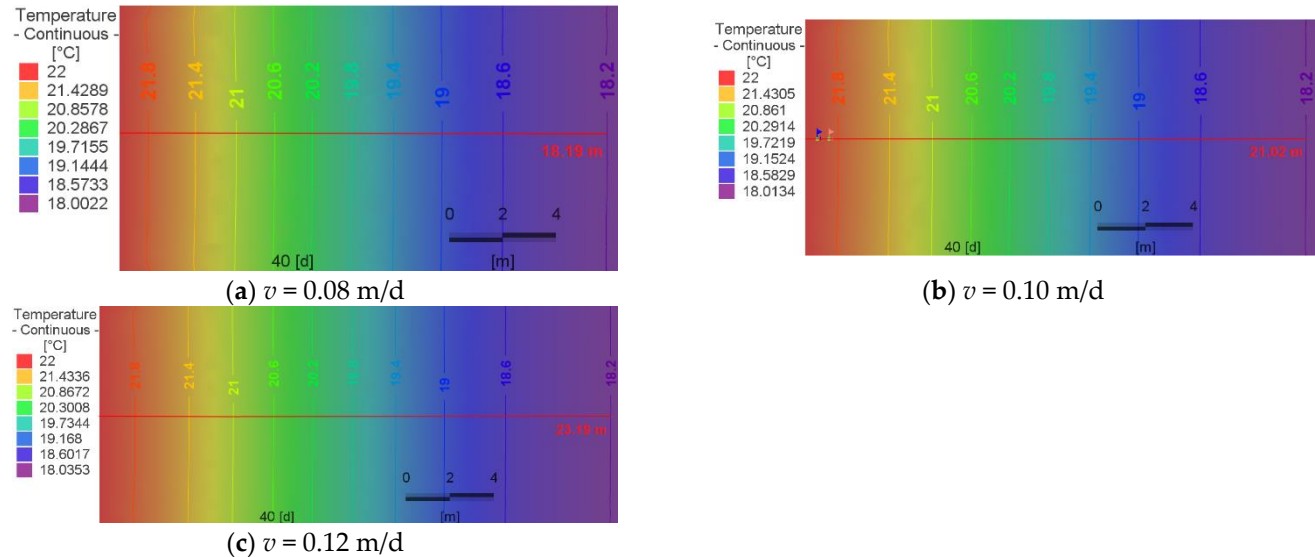

**Figure 7.** The temperature distribution of the aquifer after 40 days of model operation.

From Figures 6 and 7, the water temperature of the aquifer is influenced by the variation of water temperature in the river canal, which decreases rapidly with the increase of distance from the river canal boundary. The linear distance from the river canal to the 18.2 °C isotherm can be taken as the river canal's influence range in this example, according to the definition of thermal penetration in heat transfer [29], where the influence amplitude less than 5% of the river canal temperature variation amplitude is considered to be the influence boundary. The results show that, the influence range of the river canal boundary under the same conditions increases from 12.02 m to 15.07 m after 20 days and increases from 18.19 m to 23.19 m after 40 days as the groundwater seepage velocity $v$ increases from 0.08 m/d to 0.12 m/d.

### 4.3. Comparison of the Results of the Analytical and Numerical Methods

For $v$ = 0.10 m/d, the temperature variation process at the virtual observation points 1 ($x$ = 0.5 m) and 2 ($x$ = 1.0 m) based on model operation results is shown in Figure 8.

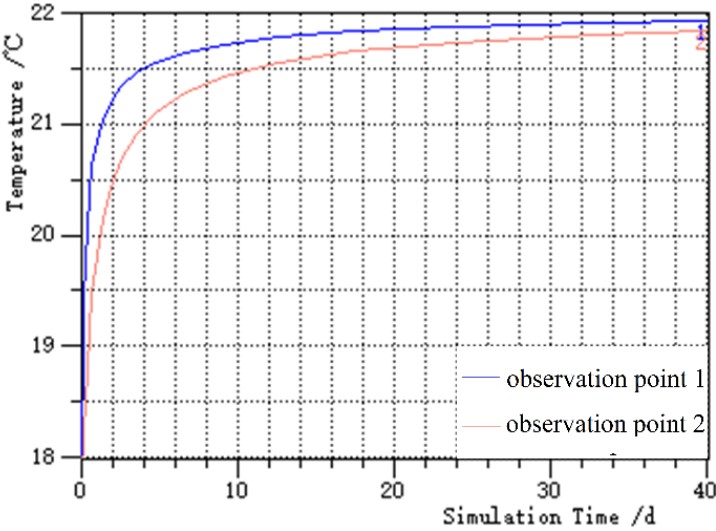

**Figure 8.** The temperature variation process of virtual observation points.

According to Figure 8, the groundwater temperature at a certain distance from the river canal boundary varies asymptotically with time.

Based on Equation (26), the excess temperature values $T_b(x,t)$ at $x$ = 0.5 m and 1.0 m are calculated, and the sum of them and the initial temperature of the aquifer (18 °C) constitute the analytic values of the corresponding aquifer temperature. Tables 2 and 3, and Figure 9, compare the outcomes of the analytical and numerical simulations, respectively.

**Table 2.** Comparison of analytical and numerical results of virtual observation point 1 ($x$ = 0.5 m).

| Time/d | Analytical Value/°C | Numerical Value/°C | Absolute Error/°C | Relative Error */% |
|--------|---------------------|--------------------|--------------------|---------------------|
| 5  | 20.7484 | 21.5715 | 0.8231 | 3.97 |
| 10 | 21.5121 | 21.7456 | 0.2335 | 1.09 |
| 15 | 21.7722 | 21.8143 | 0.0421 | 0.19 |
| 20 | 21.8836 | 21.8565 | 0.0271 | 0.12 |
| 25 | 21.9372 | 21.8830 | 0.0542 | 0.25 |
| 30 | 21.9648 | 21.9020 | 0.0628 | 0.29 |
| 40 | 21.9881 | 21.9280 | 0.0601 | 0.27 |

* The relative error is the ratio of the absolute error to the analytical value.

**Table 3.** Comparison of analytical and numerical results of virtual observation point 2 ($x$ = 1.0 m).

| Time/d | Analytical Value/°C | Numerical Value/°C | Absolute Error/°C | Relative Error */% |
|--------|---------------------|--------------------|--------------------|---------------------|
| 5  | 19.1558 | 21.1240 | 1.9682 | 10.27 |
| 10 | 20.5766 | 21.4733 | 0.8967 | 4.36 |
| 15 | 21.2641 | 21.6146 | 0.3505 | 1.65 |
| 20 | 21.6022 | 21.7014 | 0.0992 | 0.46 |
| 25 | 21.7774 | 21.7563 | 0.0211 | 0.10 |
| 30 | 21.8721 | 21.7959 | 0.0762 | 0.35 |
| 40 | 21.9553 | 21.8499 | 0.1054 | 0.48 |

* The relative error is the ratio of the absolute error to the analytical value.

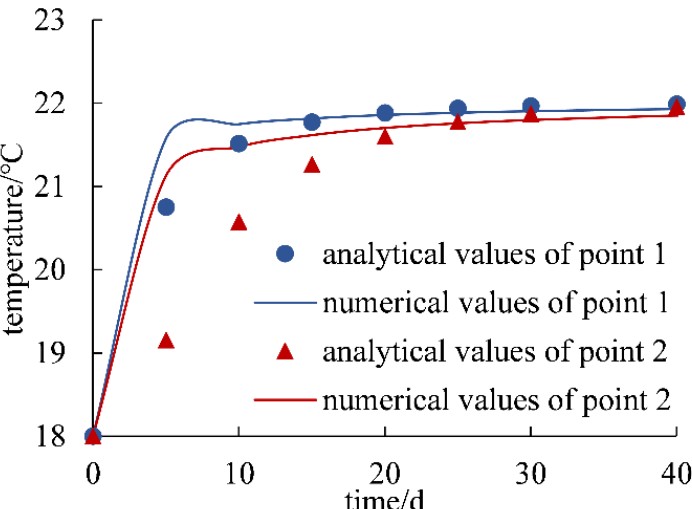

**Figure 9.** The comparison of analytical and numerical results.

From Tables 2 and 3, and Figure 9, the temperature values and their variation trends obtained by the analytical method and the numerical method are consistent, and the relative calculation errors are basically less than 5%, which validates the correctness of the analytical solution of the one-dimensional thermal convection-conduction model of the aquifer, i.e., Equation (26).

## 5. Discussion

### 5.1. Error Analysis between Analytical Results and Numerical Results

When comparing the results of the analytical method and those of the numerical method, it must be pointed out that the relative errors between the numerical and analytical values share the following laws:

1. At the same distance from the boundary, the relative error tends to gradually decrease and then slightly increase with time, and the maximum error of the two methods reaches 10.27% at the virtual observation point 2, which is relatively far from the boundary, under the condition of shorter time (such as 5 days in the example).
2. Under the condition of the same time, the closer the distance to the boundary, the relative error tends to decrease.
3. When the duration of the boundary effect is relatively short (e.g., $t < 15$ d), the comparison of the results generally shows that the numerical values are larger than the corresponding analytical values.

These are mainly because of the influence of the truncation error of the numerical calculation. When the boundary action time is short or the calculation point is far from the boundary, that is, the aquifer temperature rise is relatively small, and the relative error is relatively large, this results in laws 1 and 2. Under the influence of the rising effect of river water temperature, the groundwater temperature distribution shows an asymptotic trend along the $x$-direction with the initial temperature as the reference, which will be less than the linear interpolation of each element in the numerical method, leading to the formation of law 3.

### 5.2. The Impact of $\Delta T_0$ on $T_b(x,t)$

In the example mentioned above, the temperature variation in the river channel caused by the thermal discharge from the power plant, $\Delta T_0$ was set to a constant value, 4 °C. However, $\Delta T_0$ is affected by factors such as the operating conditions of the power plant and the seasons [30]. Therefore, it is necessary to discuss the impact of the temperature variation in the river channel on the aquifer temperature, i.e., the relationship between $\Delta T_0$ and $T_b(x,t)$.

Here, we set $a = 0.0336 \text{ m}^2/\text{d}$, $v = 0.1 \text{ m}/\text{d}$, and $t = 40 \text{ d}$. Then $T_b(x,t)$ is a linear function of $\Delta T_0$ with a slope of 0.9888 (Figure 10), according to Equation (26).

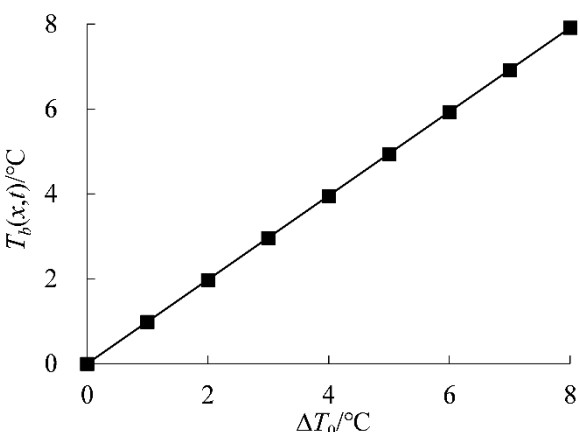

**Figure 10.** The relationship between $T_b(x,t)$ and $\Delta T_0$.

*5.3. Contributions to the Management of Rivers and Aquifers*

Groundwater temperature is an important factor affecting water quality and aquifer ecosystems and is also the basis for the development and utilization of shallow geothermal energy. Therefore, the study of the unsteady heat transfer model of groundwater temperature near the river channel is not only the basic theoretical component of systematic water use planning and management under the interaction of rivers and groundwater, but it also provides the fundamental technical basis for the development, utilization, and management of geothermal energy in the aquifer near the river channel.

## 6. Conclusions

For the one-dimensional thermal convection-conduction problem in the horizontal aquifer under the boundary condition that the water temperature rises instantaneously and remains constant in the river channel, the Cole–Hopf transformation is used in this paper to transform this problem into a one-dimensional heat conduction model under exponential function type boundary conditions, and the characteristics of the Laplace transformation are utilized to provide the analytical solution of the model. Following the analysis of aquifer temperature variation laws based on the analytical solution and numerical simulation analysis of the influence range of the boundary, the analytical solution is further confirmed, and the following main conclusions are drawn.

(1) Groundwater temperature variation in the confined aquifer under the influence of water temperature change in the river channel is controlled by the combined action of heat conduction in the aquifer medium and thermal convection formed by groundwater seepage, in which thermal convection plays a dominant role and the degree of thermal convection is proportional to the groundwater seepage velocity.

(2) Under the condition that the water temperature in the river channel rises instantaneously by $\Delta T_0$ and remains constant, the groundwater temperature at a certain distance from the river channel varies asymptotically with time. In the initial stage when the variation of groundwater temperature is relatively small, the actual amplitude of groundwater temperature variation is smaller than the linear interpolation value in each calculation element of the numerical method, which leads to the numerical value being larger than the analytical value.

(3) The results of the analytical method and the numerical method are essentially consistent, and the relative error is basically less than 5%, which verifies the accuracy of the analytical solution.

(4) The method of deriving the analytical solution of the thermal convection-conduction equation in the paper is relatively convenient, so it can also be used in solving relative

convection-dispersion and convection-diffusion problems where the Burgers' equation serves as the basic theoretical equation.

**Author Contributions:** Conceptualization, T.W. and Y.T.; methodology, T.W. and H.R.; validation, T.W.; formal analysis, T.W. and Y.T.; investigation, T.W.; resources, Y.T.; data curation, T.W.; writing—original draft preparation, T.W.; writing—review and editing, Y.T.; visualization, T.W.; supervision, Y.T.; project administration, F.L.; funding acquisition, F.L. All authors have read and agreed to the published version of the manuscript.

**Funding:** This research was funded by the Open Research Fund Program of State Key Laboratory of Hydroscience and Engineering, Tsinghua University, grant number sklhse-2020-D-06.

**Data Availability Statement:** Not applicable.

**Conflicts of Interest:** The authors declare no conflict of interest.

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
