# Peer review of "The Analytical Solution of an Unsteady State Heat Transfer Model for the Confined Aquifer under the Influence of Water Temperature Variation in the River Channel"

_water, doi:10.3390/w14223698_

Round 1
Reviewer 1 Report
The authors present a study of Unsteady State Heat Transfer Model and its Solution in the 2 Confined Aquifer near the River Channel .
In my opinion, this paper should be rewritten.
The methodology has no novel contribution? ; authors should better point out novelty of the study.
In the section of Material and Methods, the authors need to describe methodology which were used in the experimental process, how FEFlown was used? What are moddeling input , calibration , validation requirments. Please reorganize and rewrite the section of the Material and Methods.
Using the model, preparing the data needed for simulation calculations, their conduct and presentation of the obtained results is not enough to be a research paper. The authors should introduce more data, analysis and discussion into this manuscript. There are many factors can affect the modelling framework, the authors need to add more to these factors. The authors should also compared to the other methods in literature. The authors should provide R2 calculation.
Reviewer 2 Report
Dear Authors,
I believe this article is more focus on mathematics rather than water, I have the following suggestion to shift the focus:
First, I would like you to state clearly the novelty of this article,
State the assumptions made and their impact on the final results, I mean the temperature in the river channel,
Enhance the introduction by including articles related to the temperature in both aquifers and river channels,
Compare your results with real data from literature,
State how this article will contribute to our knowledge of managing river channels/aquifers etc,
Reviewer 3 Report
This manuscript, water-2011373-peer-review-v1- entitled "Unsteady State Heat Transfer Model and its Solution in the Confined Aquifer near the River Channel," is well written and has potential, but it should be more organized. This research investigates the Cole-Hopf transform was used to convert the second-order nonlinear thermal convection-conduction equation into a heat conduction equation with exponential function-type boundary conditions.
In my opinion, a careful revision of the English language should be carried out as there currently are some unclear sentences. The study seems to be well-designed. The methodology and results are technically sound. Discussions on the scientific and practical values of the study, the limitations of proposed models, and future work are meaningful. I recommend accepting this manuscript after revision. The main concerns are as follows:
1) The title should be edited and rewritten since it is too general
2) The first paragraph should explain more about the importance of modeling and groundwater problems.
3) Some abbreviations in the paper have already not been addressed in the text, such as FEFLOW.
4) More recent references might support the first and second paragraphs of the introduction. There is one research reference for 2021 and 2022. The authors should read and use the newly published papers in their research, specifically similar research worldwide.
5) More literature review about the other methods is needed. The manuscript could be substantially improved by comparing numerical, analytical, and machine-learning methods.
Vadiati, M., Rajabi Yami, Z., Eskandari, E., Nakhaei, M., & Kisi, O. (2022). Application of artificial intelligence models for prediction of groundwater level fluctuations: Case study (Tehran-Karaj alluvial aquifer). Environmental Monitoring and Assessment, 194(9), 1-21.
Samani, S., Vadiati, M., Azizi, F., Zamani, E., & Kisi, O. (2022). Groundwater Level Simulation Using Soft Computing Methods with Emphasis on Major Meteorological Components. Water Resources Management, 36(10), 3627-3647.
6) For readers to quickly catch your contribution, it would be better to highlight significant difficulties and challenges and your original achievements to overcome them more straightforwardly in the abstract and introduction.
7) Providing a comprehensive flowchart is highly recommended by researchers, so please add a flowchart representing the methodology in the paper.
8) Please provide all software used in this study. More explanation about the WolframAlpha is needed.
9) Tab. 2 and 3 are the most important table in the manuscript, and, unfortunately, the authors did not try to discuss them in a specific way. A comprehensive discussion emphasizing this would significantly improve the paper on the table.
10) The discussion section in the present form is relatively weak and should be strengthened with more details and justifications.
11) It seems that conclusions are observations only, and the manuscript needs thorough checking for explanations given for results. The authors should interpret more precisely the results argument.
Round 2
Reviewer 1 Report
Now manuscript could be accepted.
One explanation for author : R-squared (R2) is an important statistical measure. A regression model represents the proportion of the difference or variance in statistical terms for a dependent variable that an independent variable or variables can explain. In short, it determines how well the data will fit the regression model.
Reviewer 2 Report
Accept